

# Functional groups of Afrotropical EPT (Ephemeroptera, Plecoptera and Trichoptera) as bioindicators of semi-urban pollution in the Tsitsa River Catchment, Eastern Cape, South Africa

Frank Chukwuzuoke Akamagwuna[1], Augustine Ovie Edegbene[1,2], Phindiwe Ntloko[1], Francis Ofurum Arimoro[3], Chika Felicitas Nnadozie[1], Dennis Junior Choruma[1] and Oghenekaro Nelson Odume[1]

[1] Institute for Water Research, Faculty of Science, Rhodes University, Makhanda, Eastern Cape, South Africa
[2] Department of Biological Sciences, Faculty of Science, Federal University of Health Sciences, Otukpo, Benue State, Nigeria
[3] Department of Animal Biology, School of Life Sciences, Federal University of Technology, Minna, Niger State, Nigeria

Corresponding authors
Frank Chukwuzuoke Akamagwuna,
akamagwunafrank@yahoo.com
Augustine Ovie Edegbene,
ovieedes@gmail.com

## ABSTRACT

We examined the distribution patterns of Ephemeroptera, Plecoptera, and Trichoptera functional feeding groups (EPT FFGs) in five streams that drain semi-urban landscapes in the Tsitsa River catchment, Eastern Cape Province of South Africa. We undertook macroinvertebrate and physicochemical analysis over four seasons between 2016 and 2017 at eight sites in three land-use categories (Sites 1, 2 and 3), representing an increasing gradient of semi-urban pollution. Five EPT FFGs (shredders, grazers/scrapers, predators, collector-gatherers and collector-filterers) were fuzzy coded and analyzed using RLQ-R (environmental characteristics of samples), L (taxa distribution across samples) and Q (species traits) and fourth-corner analyses. Physicochemical variables, including phosphate-phosphorus, total inorganic nitrogen and temperature, were the most influential variables that significantly influenced the distribution patterns of EPT FFGs in the Tsitsa River. RLQ and the fourth-corner model revealed varying responses of FFGs to semi-urban pollution. Of the five FFGs, collectors were the most abundant EPT FFGs in the study area, exhibiting disparate responses to disturbances, with collector-gatherers associated with impacted sites and significantly associated with phosphate-phosphorus. On the other hand, collector-filterers decreased with increasing semi-urban disturbance and exhibited a significant negative association with phosphate-phosphorus, total inorganic nitrogen and temperature. Overall, this study provides further insights into the environmental factors that influence the distribution patterns of FFGs in Afrotropical streams and the potential use of FFGs as indicators of anthropogenic pollution in tropical streams and rivers.

## INTRODUCTION

Anthropogenic land-use activities, including agriculture, urbanization, and industrialization, are implicated in impairing freshwater ecosystem health and functioning worldwide (*Donoghue et al., 2013*; *Peng et al., 2020*). Further, it has been reported lately in several quarters globally, that freshwater biodiversity is reducing drastically resulting from multiple pressures such as climate change, species invasion, and channel alteration among others anthropogenic pressures (*Garcia-Giron et al., 2021*; *Ge et al., 2022*; *Reid et al., 2019*). Among the anthropogenic pressures that impact negatively on freshwater structural and functional ecology, locally induced pressures (*e.g.*, semi-urban pollution) is implicated to be the most human induced pressure that pattern the taxonomic and functional diversity of organisms. In semi-urban catchments, land use activities constantly undergo rapid changes from rural to highly urbanized landscapes due to rapid population growth and consequent rural–urban migration (*Akamagwuna et al., 2021*). These changes are characterized by subsistence and commercial agriculture, rapidly changing informal settlements, and industrial advancements, which often threaten the ecological health of freshwater ecosystems globally (*Vörösmarty et al., 2010*). For instance, land-use activities alter streams' and river ecosystems' hydrological features and water chemistry, ultimately changing the complex biotic and abiotic processes (*e.g.*, nutrient cycling, primary productivity and trophic dynamics) that shape streams' biological communities and ecosystem function (*Garcia-Raventos et al., 2017*; *Lidman et al., 2017*). This problem is expected to increase geometrically in regions such as Africa, where population growth is projected to double by 2050 (*World Bank Group, 2020*), leading to rapidly changing rural–urban landscapes. However, the critical environmental variables associated with semi-urban activities driving benthic communities and their functional organization remain unclear, especially in developing countries. For example, *Misaki, Yokomizo & Tanaka (2019)* have found varying responses of macroinvertebrate feeding groups neonicotinoid insecticides in different regions in Japan. In the African region, agricultural practices are poorly managed, and informal settlements are predominant, leading to excessive input of contaminants such as pesticides, heavy metals and nitrates in streams.

Land-use-induced environmental variables can significantly influence aquatic macroinvertebrates' functional roles (*e.g.*, food web and trophic interactions) by altering their functional organization (*Fu et al., 2016*). For example, the destruction of riparian vegetation for agricultural and informal settlements can reduce canopy cover, increase light exposure, and promote primary algal production (*Edegbene et al., 2022*; *Richoux, Moyo & Dalu, 2018*; *Seger et al., 2012*). Further, hydrological and habitat alteration can influence the input of allochthonous basal resources to rivers, affecting local macroinvertebrates' food availability. Such shifts in land use can significantly affect basal resources such as algae and detritus that support macroinvertebrate FFGs such as grazers and shredders (*Mangadze et al., 2019*; *Miserendino & Masi, 2010*; *Vilenica et al., 2020*). Similarly, fertilizer use for agriculture, detergent for clothes washing, and animal and human waste can promote nutrient inputs, favouring macroinvertebrate FFGs (*e.g.*, caddisfly; gatherers) that depend on fine particulate organic matter (FPOM) as food (*Brasil et al., 2013*; *Moyo & Richoux,*

*2018*). On the other hand, fine particles from nutrient and sedimentation inputs can cause clogging and abrasive effects on feeding nets of filter-feeding and soft and exposed body macroinvertebrates (*Akamagwuna et al., 2019*; *Jones et al., 2012*), as well as reduce the food quality for gatherers and scrapers (*Graham, 1990*). The shift in physicochemical characteristics (*e.g.*, temperature and pH) resulting from land uses may also affect leaf microbial degradation by shredders, affecting FPOM for filter-feeders. Thus, FFGs are considered reliable proxies of ecosystem functioning following a natural environmental perturbation (*Bêche & Statzner, 2009*), and human-induced disturbances (*Sitati et al., 2021*). However, the response of Afrotropical macroinvertebrate functional organization to environmental factors associated with semi-urban pollution have not been thoroughly examined, limiting our understanding and the development of FFG classification unique to the region (*Akamagwuna & Odume, 2020*; *Edegbene et al., 2022*; *Masese & Raburu, 2017*; *Sitati et al., 2021*).

Globally, macroinvertebrates taxa in the Orders Ephemeroptera, Plecoptera and Trichoptera (EPT) together with other orders have been widely explored in developing biomonitoring tools for freshwater ecosystem health (*Edegbene et al., 2021*; *Ge et al., 2022*). However, taxa of the EPTs are one of the most sensitive to anthropogenic pressures. Identification of the macroinvertebrates to genus and species levels is common in developed countries due to wide range of experts and taxonomic keys/guides. Moreover, resource scarcity and taxonomic expertise are among the major challenges faced by biomonitoring studies using macroinvertebrates in Africa (*Edegbene et al., 2021*; *Mangadze et al., 2019*; *Sitati et al., 2021*). These challenges have led to ambiguous outcomes and results of biomonitoring studies, especially when using the whole macroinvertebrate community (*Edegbene et al., 2021*; *Akamagwuna & Odume, 2020*), thus hindering the development of biomonitoring tools at genus and species levels and the use of functional indicators such as FFGs as indicators of ecosystem health in the region. In the present study, we opted to use the FFGs of the EPT, owing to the fact that the EPTs are the most abundantly distributed and diverse macroinvertebrate orders in the Afrotropical region (*Akamagwuna, 2021*). The EPTs represent Africa's most widely studied invertebrate groups; thus, considerable taxonomic and autecological information exists for FFG classification at the species level, especially in South Africa (*Akamagwuna et al., 2021*). Further, the EPT has broad sensitivity to disturbance (*Ge et al., 2022*; *Masese & Raburu, 2017*; *Pollard & Yuan, 2010*), including natural environmental changes (*Schmera, Baur & Erős, 2012*) and human-driven disturbances (*Lange, Townsend & Matthaei, 2014*). Given that all macroinvertebrate FFGs are fully represented within the EPT orders and respond differently to environmental changes, we examined the effects of semi-urban induced environmental changes on the functional organization of EPT in the Tsitsa River catchment, South Africa. Additionally, the EPT have the potential to provide details and subtle assessment of anthropogenic disturbance, including local pressures (*e.g.*, semi-urban pollution) effects on freshwater ecosystems. We also opted for the EPT for the assessing the effects of semi-urban pollution on the Tsitsa River catchment, South Africa, as most of the functional ecological studies conducted in the Afrotropic employed the use of the entire macroinvertebrates captured

in the course of the studies (*e.g.*, *Edegbene, Arimoro & Odume, 2020a*; *Edegbene, Arimoro & Odume, 2020b*; *Edegbene et al., 2022*; *Odume, 2020*; *Sitati et al., 2021*).

We expect to identify critical environmental variables associated with semi-urban disturbance influencing the functional organization of EPT and the food web of the Tsitsa River catchment. We raised specific hypotheses regarding the effects of ecological changes on FFGs. We predicted that the proportion of collector-filterers, shredders, and predators would decrease with changes in physicochemical variables such as turbidity, nutrients, and temperature across semi-urban disturbance gradients. On the other hand, collectors-gatherers and scrapers/grazers were predicted to increase along the increasing gradient of semi-urban pollution. We hope that the outcomes from this study can contribute to our understanding functional organization and food web of Afrotropical River systems and stimulate the development of cheap and local biomonitoring tools based on FFGs unique to the region.

## MATERIALS & METHODS

### Study area

The Tsitsa River catchment is 90 km long. The Tsitsa River catchment originates from the eastern escarpment of the Drakensberg Mountains near the town of Matatiele, and runs through a catchment area of 4,924 km$^2$ where it delivers into the Indian Ocean on the southeast coast of South Africa after joining the Mzimvubu River (*Theron et al., 2021*) (Fig. 1). The Tsitsa River catchment consists of three tributaries: millstream Qurana and Pot, and Little Pot Rivers. Annual rainfall in the Tsitsa catchment range between 700 mm and 1,000 mm, with the highest values occurring mainly during summer (*Akamagwuna, 2018*). The climate in the Tsitsa River catchment ranges from temperate in the northern altitude to sub-tropical along the coastal belt. Temperature peaks in January with a monthly average of ∼20 °C, and the coldest month is July, with a monthly average temperature of ∼0 °C (*Huchzermeyer et al., 2019*). The area's geology is classified as the erodible Beaufort sandstones (shales, basalt material, and alluvial deposits). The erodible nature of geology and agricultural activities contributes significantly to water quality deterioration in the area. This river flows through agricultural land, where water is extensively withdrawn for irrigation and domestic purposes. Farming in the Tsitsa catchment consists mainly of animal, vegetable, and wood production. The *Acacia* sp., including grassland and savanna bioregions, are the main vegetation in the area (*Libala, Palmer & Odume, 2020*).

### Site classification and land-use types

Eight sites were selected in five streams in the Tsitsa River catchment and classified into three site categories according to *Akamagwuna et al. (2021)* to examine the functional organization of EPT in this study. Given those semi-urban activities, including informal settlements, farming practices (livestock grazing and crop production), and industrial advancement characterize the catchments, we carefully selected sites across the five streams to represent and increasing gradient of semi-urban pollution. These land uses are spatially spread across the catchments, even across each stream, so it was expected that sites selected in different location in one stream will be independent statistically. Two sites were selected
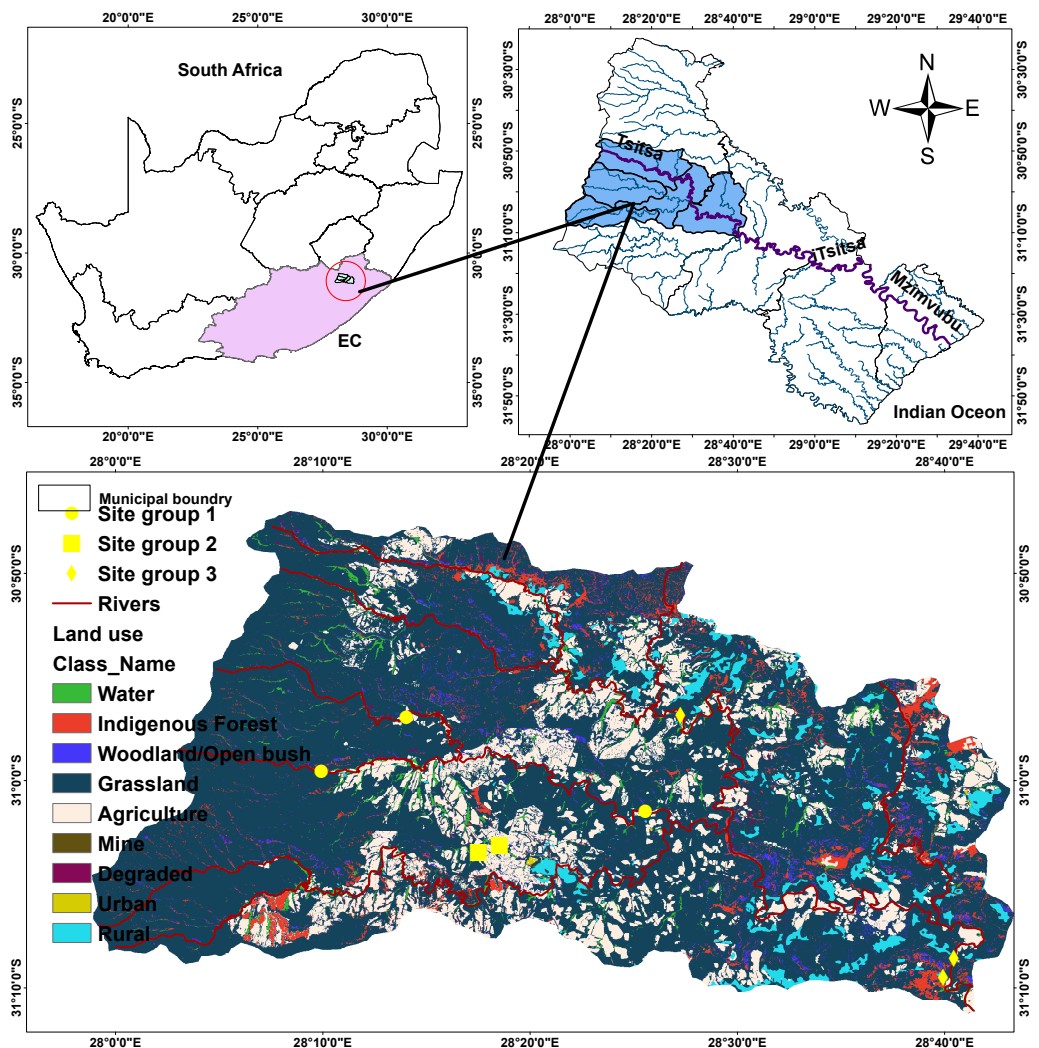

**Figure 1** Map of the study area showing sampling points within five streams in the Tsitsa River.

in the main Tsitsa River catchment channel (Tsitsa up and down streams), two sites in the Millstream (Millstream up and down), and two sites in the Pot River (Pot up and down streams). We selected one site each from the Qurana and Little Pot Rivers. We screened the land-use cover (%) for the eight sites using the South Africa land-use data map in ArcGis version 10.2 (Esri, Redlands, CA, USA), according to *Akamagwuna et al. (2021)*.

Briefly, we extracted the land-use area that drain each site and converted it to percentage land cover areas. The land covers were classified into the forest, pasture, agriculture, and settlements land covers. We classified the eight sites into three site categories using these data (Fig. 1; see *Akamagwuna et al., 2021*). The three groups include Site group 1 (Pot River: up and down, and Little Pot River; least impacted), Site group 2 (Millstream up and down; moderately impacted), and Site group 3 (Qurana tributary, and Tsitsa River up and down; highly impacted). The site categories represent an increasing gradient of ecological
conditions. Site groups 3 (highly impacted) and 1 (least impacted) were drained by >70% human settlements and forest lands, respectively. On the other hand, a combination of rural settlement and forests drain Site group 2 (moderately impacted; see *Akamagwuna et al., 2021*).

## EPT sampling, identification, and FFG classification

We collected macroinvertebrate samples from the eight sites in five streams during four sampling events: August 2016 (early winter), October 2016 (spring), January 2017 (summer), and April 2017 (autumn). We used the D-frame kick net (30 cm $\times$ 30 cm, 1,000 $\mu$m mesh size) to collect macroinvertebrates (South African Scoring System version 5, SASS 5; *Dickens & Graham, 2002*). Macroinvertebrates were kick-sampled from three disparate biotopes, including stone, vegetation and sediments. Stone biotope is defined as pebbles and cobbles (2–25 cm), and boulders (>25 cm) situated in riffles (stones in currents) and pools (stones out of current). Vegetation biotopes comprised of marginal vegetation growing on the river edge and aquatic vegetation that are submerged in water. Sediment biotope included gravels (small stones that are <2 cm in diameter), sand and mud that are <than two mm and 0.06 mm, respectively. The stone and sediment biotopes were sampled for 3 min and 1 min, respectively. Marginal and aquatic vegetation sampled were about 2 m and 1 m$^2$ of, respectively (*Dickens & Graham, 2002*). Three replicate samples were collected from each of the three biotopes per site during every sampling event. Only the dominant biotopes present at each site were sampled during each sampling event. We used 70% alcohol to preserve macroinvertebrates and transported them to the laboratory for sorting, enumeration and identification.

We first identified all macroinvertebrate groups into their respective families using taxonomic keys (*Mattson, 1996*; *De Moor, Day & De Moor, 2003a*), and (*De Moor, Day & De Moor, 2003b*). Second, we separated the EPT specimens and identified them into genus or species levels using a dissecting microscope ($\times$10 $vs$ $\times$40) with relevant keys (*De Moor, Day & De Moor, 2003a*). The identified EPT taxa were assigned into five FFGs, including scraper/grazer, shredders, predators, collector-gathers, and collector-filterers, following Merrit et al., 2008. A fuzzy coding method was used to assign FFGs to EPT taxa. We raise specific predictions regarding FFG response to environmental changes associated with semi-urban pollution (Table 1).

## Water sampling and analysis

We measured physicochemical variables concurrently with macroinvertebrate sampling across the eight sites. Variables measured using the Hanna meter includes pH, electrical conductivity (EC, mS/m), dissolved oxygen (DO; mg/l), temperature (°C) and turbidity (NTU) with turbidity meter (model 966). We collected water samples and analyzed them for nitrate-nitrogen (NO$_3$-N; mg/l) and nitrite-nitrogen (NO$_2$-N; mg/l) using the Biotek microplate and orthophosphate-phosphorus (PO$_4$-P; mg/l) with Merck spectroquant$^{\circledR}$ Phosphate test kit. We used ammonium test kits to measure ammonium-nitrogen (NH$_4$-N; mg/l), whereas total inorganic nitrogen (TIN; mg/l) was obtained according to *Akamagwuna et al. (2021)*, by totaling the nitrogen components of NO$_3$-N, NO$_2$-N,

**Table 1** Ephemeroptera, Plecoptera, and Trichoptera functional feeding groups and predicted response to semi-urban disturbance stress in the Tsitsa River, Eastern Cape, South Africa.

| EPT FFG | Predicted responses | Rationale |
|---|---|---|
| Collector-gatherers | + | Increased particulate organic matter resulting from organic inputs |
| Scraper/ grazer | + | Increased algal production due to increased temperature via riparian destruction |
| Collector-filterers | − | Sensitivity to clogging effects of fine particles |
| Shredders | − | Sensitivity decreased leaf litter input due to riparian fragmentation |
| Predators | − | Sensitivity to visual impairment resulting from organic input and turbidity in suspension |

**Notes.**
− and + indicate decreased and increased abundance, respectively (*Akamagwuna & Odume, 2020*; *Edegbene et al., 2022*; *Masese & Raburu, 2017*; *Masese et al., 2014*; *Sitati et al., 2021*).

and $PO_4$-P. All nutrient measurements followed the American Public Health Association methods.

## Data and statistical analyses

One-way analysis of variance (ANOVA) tested the difference in each physiochemical variable between the site groups, and two-way multivariate analysis of variance (MANOVA) further tested the effect of site and season on environmental variables and relative abundances of EPT FFGs. Before ANOVA and MANOVA, we employed the Kolmogorov's and Levene's tests to examine the datasets normality and equality of variance, respectively. *Post hoc* multiple comparison tests indicated site groups and seasons that differed significantly. Principal component analysis (PCA) was undertaken to show the relationships between sampling sites and physicochemical variables.

Multivariate RLQ analysis was conducted to explore the distribution structure of EPT FFGs in relation to site groups and the analyzed physicochemical factors. The RLQ analysis is a four-step method developed by *Dolédec et al. (1996)* routinely used to link species traits (*e.g.*, FFGs and life-history traits) to environmental variables using the species abundance data. Thus, the RLQ analysis is routinely conducted on three datasets, *i.e.*, the macroinvertebrate taxa L-table, physicochemical data R-table, and trait attributes and, in this case, FFGs only Q-table.

Correspondence analysis was first conducted on the L-table taxa dataset, followed by a PCA on the physicochemical variable's dataset (*i.e.*, R-table). The PCA ordination connects the taxa dataset to the environmental variable's dataset using the CA's sample scores as row weights. The third ordination, Hill-smith analysis (HS) was then conducted on traits (the FFGs) dataset (Q-table), which links the taxa dataset to the trait (FFGs) dataset with the results CA's taxon scores as row weights. The last step of the RLQ procedure simultaneously undertakes ordination analysis on the three separate ordinations (*Dolédec et al., 1996*). We used the fuzzy coded EPT FFGs dataset to represent the trait (FFGs) dataset (Q-table) for the RLQ analysis. The taxa abundance dataset, L-table used was the EPT species collected from three biotopes across the eight sites and four seasons. To assess the sensitivity and tolerance

of each FFG to physicochemical variables, we used the multivariate combined fourth-corner analysis (*Dray et al., 2014*; *Dray & Legendre, 2008*). The fourth-corner analysis shows the positive and negative correlations between individual FFGs and physicochemical variables (*Edegbene et al., 2021*). This analysis was also used to confirm and identify physicochemical variables that significantly affected FFGs in the RLQ model. This approach of identifying signature traits, including FFGs, has been successfully used in previous studies to identify trait indicators of urban pollution (*Edegbene et al., 2021*; *Odume, 2020*). The Monte-Carlo model tested the significance of the RLQ and fourth-corner analyses. Model 6 (*Dray et al., 2014*) was used. The model produces two *p*-values (*i.e.*, from Models 2 and 4) was used for the significance test. For the RLQ, 999 permutations was used, and 4,999 permutations were used for the fourth-corner model. The false discovery rate method for *p*-value adjustment was applied. We conducted all the multivariate ordinations, including RLQ and associated models, fourth-corner in R software using *ADE-4* packages (*Oksanen et al., 2019*; *R Core Team, 2020*). ANOVA, MANOVA, Levene's and Kolmogorov's tests were conducted using Statistica v10 (StatSoft, Tulsa, OK, USA).

## RESULTS

### Physicochemical variables

The mean, standard deviation and range of physicochemical variables are presented in Table 2. The MANOVA used to test the statistical differences in physicochemical variables between the site groups ($df = 18$, $F = 5.58$, $p < 0.001$ and seasons ($df = 27$, $F = 13.38$, $P < 0.001$) showed that sites and seasons significantly influenced the river's water chemistry, with Site groups 1 and 3 differing significantly (Table 2). The interaction between sites and seasons also significantly influenced the water chemistry of the Tsitsa River catchment ($df = 54$, $F = 2.527$, $P < 0.001$; Table 3). These influences were also revealed by the PCA ordination results, with Site group 3 during the summer season experiencing high water quality degradation, with increased nutrient concentrations and depleted DO values compared with Site groups 1 and 2 during winter (Fig. 2). Site groups 1 and 2 situated in Pot and Little Pot Rivers, and Mill streams, which are situated in protected farmland were less impacted during the winter than the spring season (Fig. 2). These site groups showed strong negative associations with increasing temperature $NO_3$-N and pH during the winter season. Similarly, DO influence the structuring of Site groups 2 and 3 during winter (Fig. 2). These site groups were negatively associated with nutrient concentrations.

Overall, the physicochemical results suggest increasing semi-urban pollution at Site group 3, especially during the dry summer period, compared with Site group 1 in the wet winter period. PCA's first and second axes with Eigenvalues 3.07 and 1.39 explained a total cumulative variance of 50%, suggesting that the PCA plot explained half of the variability in the relationships between the sites and the analyzed physicochemical variables (Table 4).

### Macroinvertebrate functional composition

We recorded 24 EPT taxa over the sampling site categories during the study period (Table S1). The relative abundance of collector–gatherers (36%) were the most common

**Table 2** Mean, standard deviation, and range of physicochemical variables collected during the study period.

| Variables | SG1 | SG2 | SG3 | F-value | P-value |
|---|---|---|---|---|---|
| EC (mS/m) | 48.5 ± 13.2 (15–60) | 64.2 ± 11.4 (50–79) | 91.1 ± 33.59 (46–175) | 9.23 | **0.001** |
| DO (mg/l) | 9.8 ± 2.6 (5.6–13.4) | 10.6 ± 4.68 (3.88–17.1) | 17.2 ± 30.98 (0.74–12.82) | 0.45 | 0.64 |
| pH | 6.9 ± 1.1 (4.5–8.2) | 7.7 ± 0.61 (6.5–8.6) | 7.2 ± 0.7 6.39–8.34) | 4.50 | **0.020** |
| Temp (C°) | 17.4 ± 6.6 (8.3–24.5) | 18.1 ± 9.14 (5.82–28) | 20.9 ± 4.98 (11.33–28) | 1.99 | 0.15 |
| $NO_3$ (mg/l) | 2.3 ± 2.6 (0.13–7.23) | 3.9 ± 3.38 (0.2–10.1) | 1.2 ± 1.54 (0.01–3.86) | 4.94 | **0.01** |
| $NO_2$ (mg/l) | 0.4 ± 0.97 (0.0025–3.6) | 0.8 ± 2.03 (0.00–0.52) | 0.3 ± 0.3 (0.0025–0.95) | 0.68 | 0.51 |
| $PO_4$ (mg/l) | 0.6 ± 1.07 (0.0025–2.85) | 1.3 ± 1.48 (0.0025–3.09) | 0.9 ± 1.53 (0.0025–3.63) | 4.11 | **0.028** |
| $NH_4$ (mg/l) | 0.2 ± 0.2 (0.0025–0.59) | 0.1 ± 0.26 (0.0025–0.62) | 0.3 ± 0.4 (0.0025–1.08) | 0.73 | 0.48 |
| TIN (mg/l) | 0.9 ± 0.077 (0.04–1.68) | 1.2 ± 1.06 (0.048–3.14) | 0.7 ± 0.56 (0.01–1.76) | 3.06 | 0.064 |

Notes.

P-values in bold showed significant difference among the site groups in terms of physicochemical variables.

**Table 3** Multivariate analysis of variance and *post hoc* results showing the significant difference between site groups and seasons in terms of physicochemical variables.

| Effect | Test | Value | F | Effect Df | Error Df | P-value |
|---|---|---|---|---|---|---|
| Intercept | Wilks | 0.000986 | 1351.535 | 9 | 12.00000 | 0.000000 |
| SG1 | Wilks | 0.037161 | 5.583 | 18 | 24.00000 | 0.000070 |
| Season | Wilks | 0.000880 | 13.381 | 27 | 35.68851 | 0.000000 |
| SG1*Season | Wilks | 0.003305 | 2.517 | 54 | 65.78235 | 0.000201 |
| *Post hoc* comparison tests. Error: Between MS = 532.90, *df* = 29.000 | | | | | | |
| Sites | SG1 | SG2 | SG3 | | | |
| SG1 | | 0.271209 | 0.000729 | | | |
| SG2 | 0.271209 | | 0.089089 | | | |
| SG3 | 0.000729 | 0.089089 | | | | |

**Table 4** Properties of principal component (PCA) results of physicochemical variables analysed during the study.

| PCA properties | PCA 1 | PCA 2 |
|---|---|---|
| Eigenvalue | 3.07 | 1.39 |
| Proportion Explained (%) | 34 | 15 |
| Cumulative Proportion (%) | 34 | 50 |

FFG in the streams, followed by collector–filterers, which were mostly abundant in the least impacted sites (31%). Shredders (5.2%) were the least abundant FFGs, followed by scrapers/grazers (6.4%) (Fig. 3). ANOVA used to analyse the effects of sites on the relative abundance of EPT FFGs showed that collectors-filterers (*df* = 31, F = 3.4, *p* = 0.04) and scrapers/grazers (*df* = 31, F = 3.98; *p* = 0.03) increased significantly in the least impacted sites compared to the highly impacted sites (F = 3.42, *p* = 0.04; Fig. 3). Conversely, shredders were more abundant in the moderate and highly impacted Site group 2 than the least impacted Site group 1, whereas collector-gatherers significantly dominated the highly impacted sites compared to the least impacted sites (*df* = 31, F = 4.964, P = 0.013; Fig. 3).

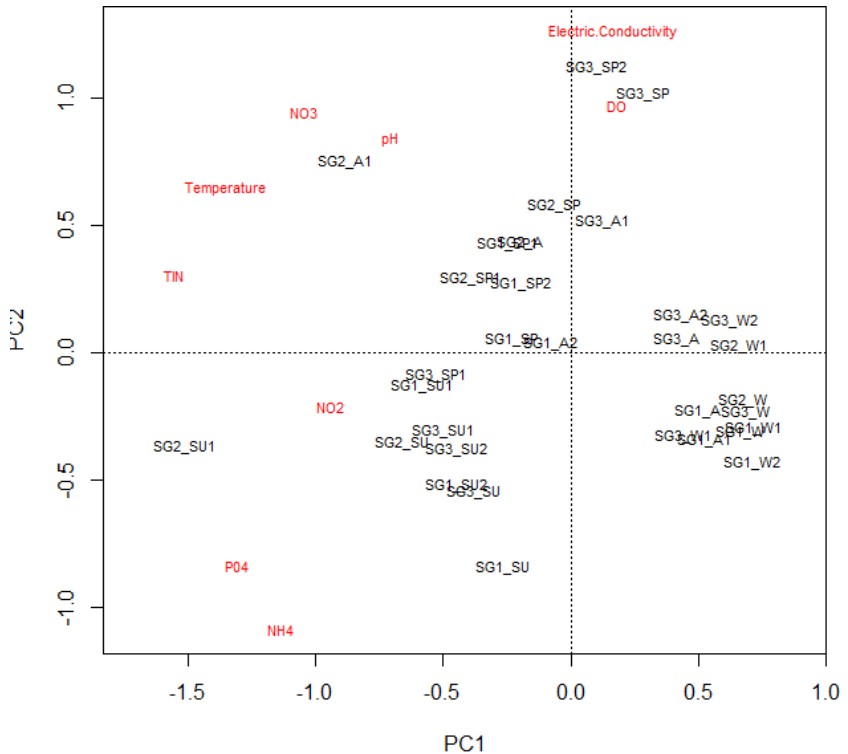

**Figure 2** **The principal component analysis plot showing the site groups' interactions and physicochemical parameters.** Site group/season of sampling: SG1 (Site group 1: Pot River up and down, and Little Pot River, least impacted sites), SG2 (Site group 2: Millstream up and down, moderately impacted sites), SG3 (Site group 3: Qurana tributary, and Tsitsa River up and down, highly impacted sites); seasons: W (winter), SP (spring), SU (summer), and A (autumn).

## Relationships between physicochemical variables and EPT FFGs

The RLQ analysis result showed that the EPT FFGs structure varied significantly between the site groups (Fig. 4). Collector-gatherers were associated with increasing semi-urban pollution at the highly impacted sites in Site group 3 and were favoured by nutrients and temperature. The occurrence of collector-gatherers in the impacted sites in Site group 3 was represented by the assemblage of *Baetis* spp., *Oligoneuropsis lawrencei* and *Aphenicerca* spp. Grazers and shredders also appeared tolerant, as semi-urban pollution increased at Site groups 2 and 3. On the other hand, collector-filterers showed positive associations with Site groups 1 and 2, mainly during spring. They were represented by *Pseudocloeon vinosum*, *P. glaucum*, *P. piscis* and *Euthraulus* spp. Shredders were represented by *Cheumatopsyche thomasetti*, *C. afra* and *Hydropsyche* spp., and showed positive associations with EC and nutrients in Site group 2 (Fig. 4). The first three RLQ axes accounted for 99% variation between the EPT feeding groups and physicochemical variables (Table 5).
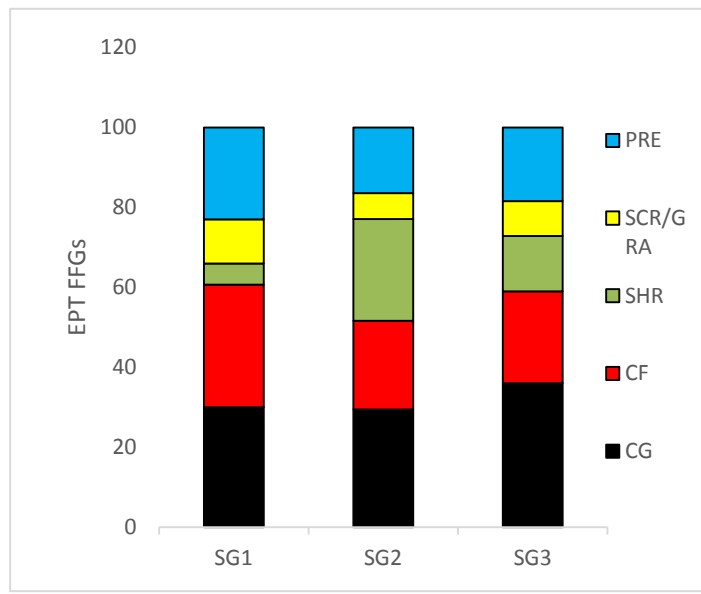

**Figure 3** **The relative abundance of Ephemeroptera Plecoptera and Trichoptera (EPT) feeding groups collected in the Tsitsa River during the study period.** SG = Site group 1 –3. Abbreviation: SCR/GRA, scraper/grazer; SHR, shredders; PRE, predators; CG, collector-gathers; and CF, collector-filterers; SG1 (Site group 1: Pot River, up and down, and Little Pot River, least impacted sites), SG2 (Site group 2: Mill-stream up and down, moderately impacted sites), SG3 (Site group 3: Qurana tributary, and Tsitsa River up and down, highly impacted sites). Different letters between site groups indicate significant differences ($p <$ 0.05) revealed by ANOVA, while similar letters indicate no significant difference between site groups ($p >$ 0.05).

**Table 5** **Properties of the RLQ method for physicochemical variables and Ephemeroptera, Plecoptera and Trichoptera functional feeding groups datasets collected in the Tsitsa River during the study period.**

| Properties (RLQ) | $P$–values: Model 2 (0.008); Model 4 (0.003) | | |
|---|---|---|---|
| | Axis 1 | Axis 2 | Axis 3 |
| Explained variance (%) | 91 | 6 | 3 |
| Cumulative variance (%) | 91 | 96 | 99 |
| Eigenvalue | 0.38 | 0.024 | 0.011 |

## Individual correlations between EPT FFGs and physicochemical variables

The combined fourth-corner analysis indicated four positive and negative correlations with EPT FFGs (Fig. 5). Three variables (TIN, $PO_4$-P, and temperature) influenced EPT FFGs, showing significant positive or negative relationships with the FFGs. For example, collector-filterers that decreased with increasing semi-urban pollution in RLQ analysis results revealed significant negative correlations with temperature, TIN, and $PO_4$-P. Collector-gatherers were also negatively correlated with the highly impacted Site group 3 in the RLQ model. On the other hand, collector-gatherers showed significant positive associations with $PO_4$-P. Collectors-gatherers were correlated with Site group 3 in the RLQ
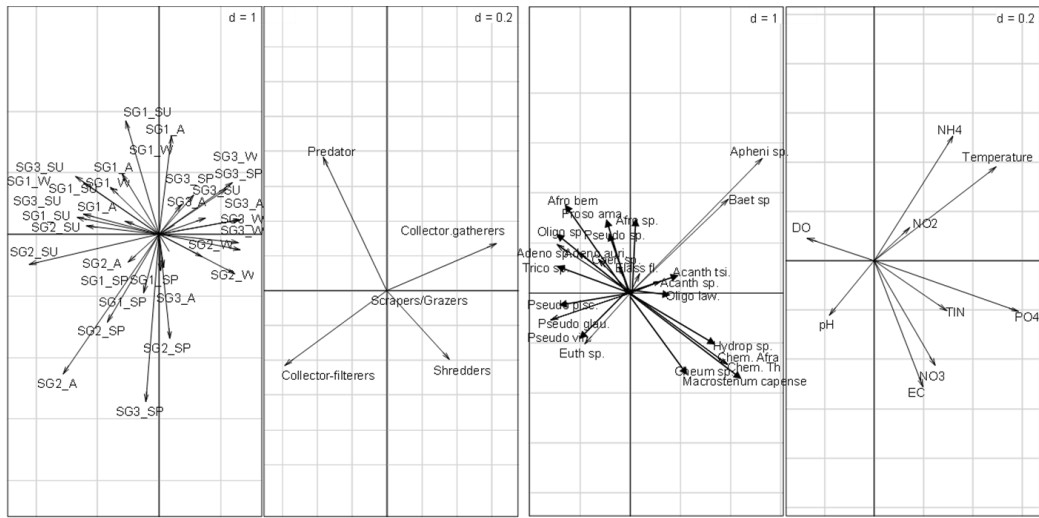

**Figure 4** **RLQ plot showing the EPT assemblages relating to site groups (A), functional feeding groups (B), and EPT abundance (C) and physicochemical variables (D).** SG1 (Site group 1: Pot River, up and down, and Little Pot River, least impacted sites), SG2 (Site group 2: Millstream up and down, moderately impacted sites), SG3 (Site group 3: Qurana tributary, and Tsitsa River up and down, highly impacted sites); seasons: W (winter), SP (spring), SU (summer), and A (autumn); Taxa: Acanth_sp (*Acanthiops*, sp.), Acanth_tsi (*Acanthiops tsitsa*), Adeno_sp (*Adenophlebia* sp.), Adeno_aur (*Adenophlebia auriculata*), Afro_sp. (*Afronurus* sp.), Afro_ber (*Afronurus bernardi*), Baet_sp (Baetis sp.), Caen_sp. (*Caenis* sp.), Pseudo_pisc (*Pseudocloeon piscis*), Pseudo_glau (*Pseudocloeon glaucum*), Pseudo_sp (*Pseudocloeon* sp.)., Pseudo_vin (*Pseudocloeon vinosum*), Apheni_sp (*Aphanicera* sp.), Cheum_sp (*Cheumatopsyche* sp.), Proso_amamz (*Prosopistoma amamzamanya*) and Hydrop_sp (*Hydropsyche* sp.); Physicochemical variables: DO (Dissolved oxygen), NH4 (ammonium), NO_3 (nitrate-nitrogen), NO_2 (nitrite-nitrogen), TIN (total inorganic nitrogen), PO_4 (phosphate-phosphorus) and EC (electrical conductivity).

analysis (Fig. 4). Overall, nutrients and temperature were the most influential variables that influenced EPT FFGs in the Tsitsa River catchment.

## DISCUSSION

This study revealed varying effects of physicochemical indicators of semi-urban disturbance on EPT FFGs, with FFGs indicating individual responses to increasing pollution. The differential responses of EPT FFGs to pollution observed in this study have been reported in North America (*Gerth et al. , 2017*), South America (*Bere, Chiyangwa & Mwedzi, 2016*; *De Castro et al., 2016*) and Africa (*Akamagwuna & Odume, 2020*; *Edegbene et al., 2022*; *Mangadze et al., 2019*) semi and urban rivers impacted by anthropogenic pollution. *Edegbene et al. (2022)* and *Sitati et al. (2021)* observed significant shift in macroinvertebrate FFGs across land-use impacted sites in Afrotropical Rivers in the Niger Delta region of Nigeria and the Kipkaren River in western Kenya, respectively. Elsewhere, (*Miserendino & Masi, 2010*) observed varying responses of macroinvertebrates FFGs to land-use disturbances in Patagonian streams. Furthermore, the RLQ analysis revealed that seasonal differences played a critical role in shaping the assemblage patterns of EPT FFGs in the Tsitsa River catchment. Seasonal heterogeneity of hydromorphological regimes is a

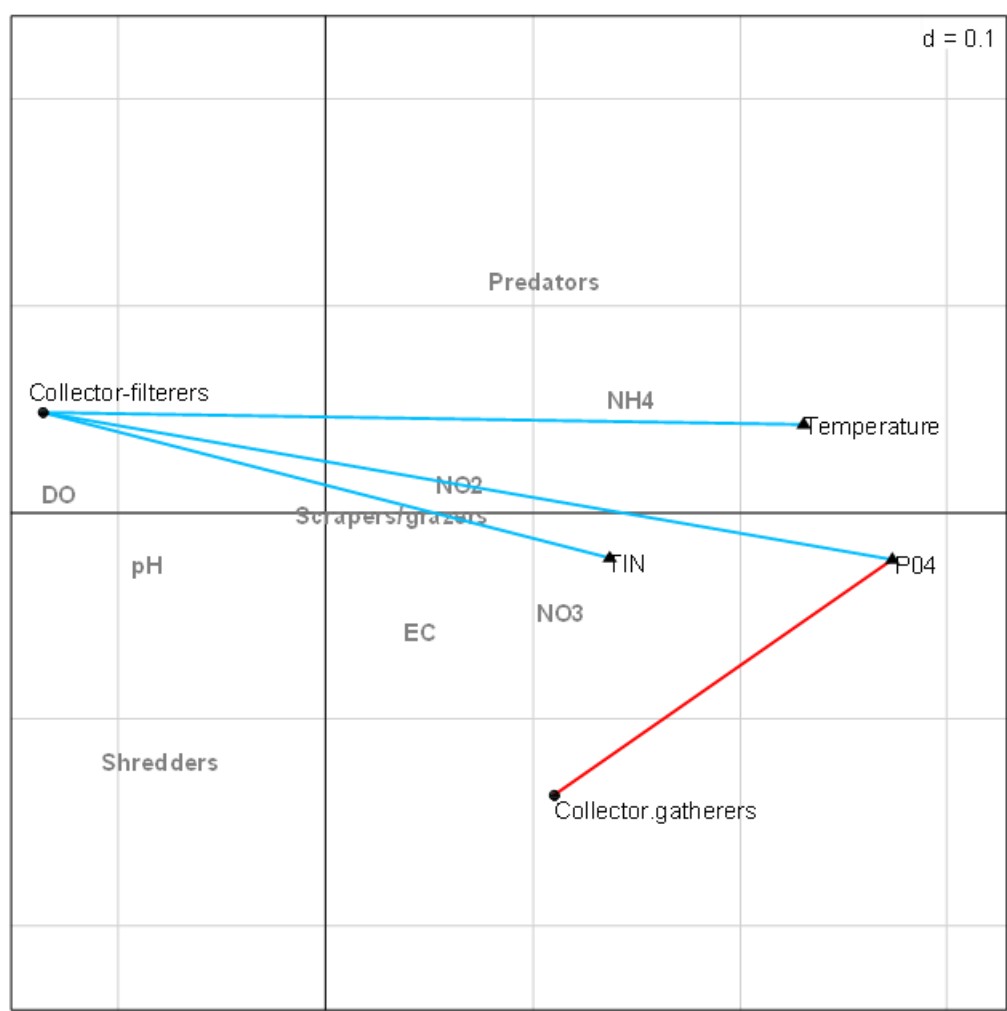

**Figure 5** **Combined fourth-corner correlation test showing the significant correlations between EPT FFGs and physicochemical indicators of semi-urban pollution in the study area.** Red- and blue-coloured lines showed significant positive and negative relationships, respectively. Grey coloured texts indicate no significant relationships between EPT FFGs and physicochemical variables ($p > 0.05$). Abbreviations: Physicochemical variables: DO (Dissolved oxygen), NH4 (ammonium), $NO_3$ (nitrate-nitrogen), $NO_2$ (nitrite-nitrogen), TIN (total inorganic nitrogen), $PO_4$ (phosphate-phosphorus) and EC (electrical conductivity).

fundamental process shaping stream communities (*Bon et al., 2021*; *Mathers, Rice & Wood, 2017*). Seasonal differences in precipitation and run-offs can influence hydrological and sedimentation dynamics in rivers, thereby affecting the aquatic community distribution. In this regard, the EPT FFGs that are highly vulnerable to clogging and abrasion by fine sediments would be severely affected (*Jones et al., 2012*). Additionally, autochthonous and allochthonous food resources input to streams changes in response to seasonal conditions, adversely affects the dynamics of FFGs distribution. These explanations may suggest the seasonal trend of EPT FFGs observed in the Tsista River during the study period. The influence of season on invertebrate assemblages have been observed in various studies in

the tropics (*Bon et al., 2021*; *Sitati et al., 2021*). *Wang et al. (2019)*, in a study to examine the effect of land use activities on the functional structure of macroinvertebrates, showed that season played a critical in influencing the effect of agriculture on the functional structure of macroinvertebrates in the Changbai Mountains in northeast China.

Our study highlights the significance of phosphate, nitrogen, and temperature in shaping the distribution patterns and functioning (*e.g.*, FFGs) of aquatic communities in rivers. Nutrients such as $PO_4$-P, TIN, and temperature were the most significant variables that influenced the distribution patterns of EPT FFGs, with collectors particularly affected by the variables, showing varying responses to semi-urban pollution (Figs. 3 and 4). For example, collector-gatherers increased with increasing pollution from semi-urban disturbance, while collector-filterers had an opposite trend in our study system. Most collectors are generalist feeders (*i.e.*, feed on a wide variety of food items) and can inhabit and thrive in varieties of stream bottom habitats, increasing their survival and reproductive potentials. This may suggest why collector-gatherers increased with increasing semi-urban pollution and positively correlated with $PO_4$-P in this study, unlike specialist feeders with limited feeding options and occupying a narrower niche. Furthermore, collector-gatherers are frequently reported as the most abundant macroinvertebrate FFGs in tropical and temperate streams because of their tolerance to nutrient pollution (*Akamagwuna et al., 2021*; *Lubanga et al., 2021*; *Mangadze et al., 2019*; *Masese et al., 2014*). Our study provides further evidence to support the high abundance of collector-gatherers in Afrotropical streams and insights into the response patterns of collectors to environmental stressors.

On the other hand, land use activities might have caused elevated sediment inputs into the highly impacted sites beyond the background level that filter-feeding macroinvertebrates can tolerate. Fine particles resulting from nutrient and sediment loading are known to clog sensitive feeding nets of collector-filterers (*Mathers, Rice & Wood, 2017*; *Wilkes et al., 2017*), suggesting why they showed an opposite trend to collector-gatherers and negatively correlated with TIN, $PO_4$-P, and temperature. Elevated nutrients from fertiliser application and increased temperature from clear-cutting of riparian vegetation for agricultural purposes and domestic activities (*e.g.*, laundry, animal, and human waste) can act together to promote algal bloom, leading to decreased DO (*Griffin, 2017*). These may significantly affect macroinvertebrate collector-filterers that are highly sensitive to organic pollution and depleting DO in water (*Bere, Chiyangwa & Mwedzi, 2016*). Although collector-filters can benefit from organic particles as food, these combined effects may suggest why collector-filterers in the Tsitsa River catchment decreased with increasing physicochemical variables as predicted. Also, collector-filterers may respond to the effects of temporal variations in hydrological and flow regimes as they require specific flow conditions to filter food particles from the water column. Both vary across seasons (*Datry et al., 2014*). These explanations further suggest the decrease in collector-filterers with increasing pollution in the Tsitsa River catchment, even though they are generalist feeders. These results are incongruent with previous studies that found significant effects of nutrients on macroinvertebrate FFGs distribution patterns in river and stream ecosystems (*Ceneviva-Bastos et al., 2017*; *Siegloch et al., 2017*; *Zhang et al., 2013*). For example, *Dalu et al. (2017)* found phosphate concentration as one of the significant variables explaining the

variation of FFGs structure across all sites in a study to understand the drivers of community structure in an Afrotropical river in the eastern Highlands of Zimbabwe. *Wang et al. (2012)* also found macroinvertebrate filters decrease significantly in urban sites, whereas collectors increase in the reference sites in the Qiangtang River, China.

Overall, the response of collectors in this study highlights the significance of nutrients and temperature in shaping stream assemblages and the utility of FFGs as indicators of anthropogenic pollution in riverine systems. For instance, the contrasting responses of collectors-gatherers and collector-filterers provided more insights into the effects of semi-urban disturbance on the functional structure of aquatic insect communities in the Tsitsa River catchment. These findings provide evidence to support existing studies in the Afrotropical region (*e.g.*, *Akamagwuna & Odume, 2020*; *Edegbene et al., 2021*; *Edegbene et al., 2022*; *Lancaster et al., 2008*; *Masese et al., 2014*) and elsewhere (*e.g.*, *Tomanova, Goitia & Helešic, 2006*; *Jiang, Xie & Chen, 2011*) that have found collectors to dominate and respond markedly to increasing anthropogenic pollution. *Miserendino & Masi (2010)* reported FFGs attributes such as collectors to be among the best indicators of land use compared to structural measures in the Patagonian streams.

## CONCLUSIONS

The FFGs of EPT responded differentially to a semi-urban disturbance in the Tsitsa River catchment. Physicochemical variables, including $PO_4$-P, TIN and temperature, were the most influential variables that shaped EPT FFGs in the catchment. The most sensitive FFGs were collector-filterers and collector-gatherers, indicating varying responses to increasing semi-urban pollution and physico-chemical variables. Phosphate, nitrogen, and temperature were the most significant nutrient variables that influenced the distribution patterns of EPT FFGs during the study period. Overall, this study provides more insights into the effects of environmental factors on macroinvertebrates' FFG distribution and their use as potential indicators of the ecosystem health of rivers and streams that drain semi-urban landscapes. Although our results indicate the potential for functional analysis of the FFGs of EPT communities to be used to monitor and evaluate anthropogenic pollution impacted by semi-urban disturbance, we recommend more studies, including more sampling sites and seasons, to validate some of the findings of the study.

### Funding

This work was supported by the Water Research Commission of South Africa (No. K1/7157), and the Rhodes University Council. The funders had no role in study design, data collection and analysis, decision to publish, or preparation of the manuscript.

### Grant Disclosures

The following grant information was disclosed by the authors:
Water Research Commission of South Africa: K1/7157.
Rhodes University Council.

## Competing Interests

The authors declare there are no competing interests.

## Author Contributions

- Frank Chukwuzuoke Akamagwuna conceived and designed the experiments, performed the experiments, analyzed the data, prepared figures and/or tables, authored or reviewed drafts of the article, and approved the final draft.
- Augustine Ovie Edegbene analyzed the data, prepared figures and/or tables, authored or reviewed drafts of the article, and approved the final draft.
- Phindiwe Ntloko performed the experiments, prepared figures and/or tables, and approved the final draft.
- Francis Ofurum Arimoro analyzed the data, prepared figures and/or tables, and approved the final draft.
- Chika Felicitas Nnadozie analyzed the data, prepared figures and/or tables, and approved the final draft.
- Dennis Junior Choruma analyzed the data, prepared figures and/or tables, and approved the final draft.
- Oghenekaro Nelson Odume conceived and designed the experiments, performed the experiments, authored or reviewed drafts of the article, and approved the final draft.

## Data Availability

The raw data is available in the Supplementary File.

## Supplemental Information

Supplemental information for this article can be found online at http://dx.doi.org/10.7717/peerj.13970#supplemental-information.

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

## FURTHER READING

**Edegbene AO, Arimoro FO, Odume ON, Ogidiaka E, Keke UN. 2021.** Can macroinvertebrate traits be explored and applied in biomonitoring riverine systems draining forested catchments? *Frontiers in Water* **3**:607556 DOI 10.3389/frwa.2021.607556.

**Edegbene AO, Adam MB, Gambo J, Osimen EC, Ikomi RB, Ogidiaka E, Omovoh GO, Akamagwuna FC. 2021.** Searching for indicator macroinvertebrate traits in an Afrotropical riverine system: implication for ecosystem biomonitoring and sustainability. *Environmental Monitoring and Assessment* **193(11)**:711 DOI 10.1007/s10661-021-09450-y.

**Poff NL, Olden JD, Vieira NKM, Finn DS, Simmons MP, Kondratieff BC. 2006.** Functional trait niches of North American lotic insects: traits-based ecological applications in light of phylogenetic relationships. *Journal of the North American Benthological Society* **25(4)**:730755 DOI 10.1899/0887-3593(2006)025[0730:FTNONA]2.0.CO;2.

**Wood P, Armitage P. 1997.** Biological effects of fine sediment in the lotic environment. *Environmental Management* **21(2)**:203–217 DOI 10.1007/s002679900019.