# Peer review of "Functional groups of Afrotropical EPT (Ephemeroptera, Plecoptera and Trichoptera) as bioindicators of semi-urban pollution in the Tsitsa River Catchment, Eastern Cape, South Africa"

_PeerJ, doi:10.7717/peerj.13970_

## Round 0.1 · original submission · Major Revisions

The manuscript has been evaluated by three reviewers. Although all three found some merit in the study, each identified a number of significant issues with the manuscript (editorial), analyses (e.g., how they were described with respect to potential reproducibility, statistical packages used), how taxa were assigned to functional groups, and the issue of pseudo replication (are samples from each of the five streams statistical replicates?). Generally speaking, the reviewers found the manuscript to be well written but identified also issues regarding clarity and suitability of figures and legends.

Reviewer 1 ·

Basic reporting

See additional comments (section 4)

Experimental design

See additional comments (section 4)

Validity of the findings

See additional comments (section 4)

Additional comments

I have reviewed the manuscript (70301) entitled “Functional groups of Afrotropical EPT (Ephemeroptera, Plecoptera and Trichoptera) as bioindicators of semi-urban pollution in the Tsitsa River, Eastern Cape, South Africa.” The manuscript is well-written and the study fills an important gap in the knowledge base about macroinvertebrate biomonitoring in urbanized Afrotropical streams. While I am not an expert in RLQ, the statistical methods appear to be sound and appropriate for the research question and data. I marked the manuscript minor revision because I have a couple of concerns and comments.

My major concern is the relatively small number of streams sampled for this study. While I appreciate the authors effort to collect 32 samples, they only sampled from five streams. There is a wide of body research on pseudoreplication that the authors should consider (Hurlberet 1984). I’m not suggesting that more data needs to be collected but I think the authors should address this concern in the methods and discussion. For example, why were only 5 streams selected? Given the effect of sample size on ANOVA and RLQ analysis, the authors should make the case for why sites sampled on the same stream and multiple times should be considered independent in the analysis. There are several papers that have argued this point and can serve as a guide (i.e. Tiemann et al 2004).

Another concern I have is related to the assignment of functional feeding group. It’s not clear to me exactly how the taxa were assigned traits. The authors cite Merrit and Poff as sources, but those traits are for North American taxa. The authors should provide a rationale for why these sources were used to assign feeding traits to taxa in Afrotropical streams, especially considering the well-known plasticity of insect feeding (Zah 2001). I understand that there may not be a regional resource for assigning FFGs, but the authors should address the use of North American traits. Another related concern is the fuzzy coding of the taxa. The authors write that they used a binary approach (line 184), then they write that the FFG traits were fuzzy coded (line 221-222, 1-5 scale in the supplemental file), but what source did they use? Given the importance of traits to this study, the authors should provide more details about FFGs were assigned.

A minor concern I have is the lack of detail about how macroinvertebrates were enumerated. Did the authors count the entire samples, or did they subsample? If they subsampled, what was the minimum counting criteria?

A minor concern is the use the of just EPT taxa. Why not Diptera, especially considering the general tolerance of Diptera to poor stream quality. The authors should provide a rationale for this decision and consider the implications of leaving the Diptera out.

Other edits
Line 31 : sites 1, 2 and 3 is a bit confusing. Unless misunderstanding, why use the word “site” to indicate a landuse category?

Citations
Hurlbert, S. H. (1984). Pseudoreplication and the design of ecological field experiments. Ecological monographs, 54(2), 187-211.
Tiemann, J S, Gillette DP, Wildhaber ML, Edds DR (2004) Effects of lowhead dams on riffle-dwelling fishes and macroinvertebrates in a midwestern river. Transactions of the American Fisheries Society, 133(3): 705-717.
Zah, R., Burgherr, P., Bernasconi, S. M., & Uehlinger, U. (2001). Stable isotope analysis of macroinvertebrates and their food sources in a glacier stream. Freshwater Biology, 46(7), 871-882.

Reviewer 2 ·

Basic reporting

Authors studied the distribution patterns of Ephemeroptera, Plecoptera, and Trichoptera belonging to the functional feeding groups (EPT FFGs) in five streams that drain semi-urban landscapes in the Tsitsa River catchment, Eastern Cape Province of South Africa. From their study alone they found the different structures of EPT organisms in different sections of rivers. The manuscript is generally well written, methods are appropriate for such a study, however, they were not described with sufficient information to be reproducible by another investigator. Literature sources were adequately cited. After carefully reading the manuscript I have a feeling that there is no novelty in Author's finding, Authors conducted previously much research on EPT groups in this area and even in the same rivers. Overall findings regardless of good study design are rather of local problems, and in that manuscript, no wider context was provided. These are the main doubts which I have about that study.
Below are some suggestions and comments on the manuscript which may help authors in their revision.
L 79: Please remove the square bracket
L: 90-107 EPT taxa and different indices were used for a long time and in many monitoring studies and as indicators of aquatic environments, especially in river systems, therefore there is no novelty here, only the use of the existing method in several rivers in the southern part of Africa.
L 140-141. The first sentence of the paragraph- please rewrite, it sounds strange
In the entire manuscript when Authors write about the Tsitsa River – the basin or catchment must be added because the readers will have a feeling that the authors studied only this river not 5 rivers in the Tsitsa River basin, e g., L107, 109,
L 160-161: eight sites in five rivers are not a big amount of the data, moreover only the EPT groups were studied, not all benthic assemblages. Other benthic invertebrates also belong to the foot web of the river system. That is why in my opinion this study is interesting but of small local value.
Data analysis, and statistics: there is no information on which statistical software was used in statistical analyses, ANOVA, MANOVA, RLQ
L 306-308 FFG and EPT response to different ranges of environmental pollution that is why they are used as a tool in monitoring studies, in the assessment of water quality and ecological status- there is no novelty here. In this study, only differences (small differences) were obtained between the three types of study sites
L 320: There is no way not to show differences in abundance, density, diversity of aquatic organisms in seasons, in each season environmental factors are changing, water parameters show different values, the temperature in the water is different and consequently, community structure, also EPT taxa change also qualitatively and quantitatively, so there is no need to explain such a thing, that's obvious.
Discussion is written in the context of the Tropical zone, African region. In my opinion, a wider context is needed and comparison with different rivers in the other parts of the world.
Conclusion: L 379-380 Authors write that study demonstrated that semi-urban pollution had varying effects on the FFGs- but in fact, these effects are very small, differences in the taxa abundance in the three types of study sites are small, and the composition of fauna are similar, polluted sites have small values of measured parameters, it is strange that from such differences the statistical significance was obtained. Rivers in the catchment can vary in terms of water flow, bottom sediments, depth, and also other factors that can have an impact on invertebrate assemblages so the results are rather weak.

Experimental design

A low number of study sites were taken into account, The context of the study is narrow. Research questions refer to a local problem, in parts are easy to predict.

Validity of the findings

The manuscript is generally well written, methods used are appropriate, overall study topic is interesting but small amount of data was used, low number of sites (local problem), and lack of the wider context of the study.

Reviewer 3 ·

Basic reporting

line 63-64: The authors said, “However, the critical environmental variables associated with semi-urban activities driving benthic communities and their functional organization remain unclear, especially in developing tropical countries.” Please explain the critical environmental variables in developed countries and compare the results in this research with other research.
Some examples in Japan:
Misaki T., H. Yokomizo, and Y. Tanaka (2019) Ecotoxicology and Environmental Safety 171, 173-180
Takeshita K.M., T.I. Hayashi, and H. Yokomizo (2020) Science of the Total Environment 140627

line 131: Could you write more precise average temperatures?

Line 299: PO5 should be PO4.

Some figures and tables should be revised because some information is not included.
Figure 1: This figure is not informative. In this figure, I do not know where Mzimvubu River, millstream, Qurana, Pot, and Little Pot Rivers are. I cannot read the legends of this figure.

Figure 4: What do the values of d in the upper right corners? Arrows should be added in (c). E.C. should be placed near an arrow in (d).

Figure 5: I cannot see gray letters on printed paper.

Tables 2: What is the range of physicochemical variables (95% Confidence Interval?)? Some ranges look strange such as “0.8 2.03 (0.00 –0.52)” and “17.2 30.98 (0.74 –12.82)” because the ranges don’t include mean.

Table 3: Wilks test should be mentioned in the main text.

Table 5: What are the values, 0.008 and 0.003?

Experimental design

'no comment'

Validity of the findings

Figure 3: I cannot see the letters showing the significant difference. It is also not clear that the difference is a difference in relative abundance or absolute abundance. Statistical test of compositional data analysis is not simple, so be careful (see Aitchison J 1982 Jour. Roy Statist. Soc. Ser. B.).

Line 203-204: Explain how you used Levene’s and Kolmogorov’s tests. Explain separately. Assumption of variance” does not make sense.

Line 211: The authors should explain fuzzy code. Explain the values in table S3.

Line 330: This sentence is not clear. What do you want to say?

Additional comments

This paper analyzed the relationship between functional groups in abundance and physicochemical indicators in the Tsitsa River, South Africa. Although the authors analyzed data in the same river, they did not focus on functional groups in EPT species. So, this paper adds new findings using the same survey.

---

## Round 0.2 · Major Revisions

As you will see reviewers 2 and 3 have remaining concerns about the manuscript that need to remedied before it is acceptable for publication. Reviewer 2 points out that a number of, in my opinion (and theirs), substantive criticisms were ignored or not adequately addressed. In particular there are elements of the manuscript that require a broader context. Also, note that reviewer 2 has questions regarding the significance of the results - although statistically significant could the differences between sites be explained by factors other than pollutants?

Reviewer 1 ·

Basic reporting

No comment

Experimental design

no Comment

Validity of the findings

no comment

Additional comments

The authors have sufficiently addressed my comments.

Reviewer 2 ·

Basic reporting

The manuscript version submitted for review is still being found as local problems, and no wider context was provided in the revision.

Experimental design

L: 93-110 was not corrected. EPT taxa and different indices were used for a long time and in many monitoring studies and as indicators of aquatic environments, especially in river systems; therefore, there is no novelty here, only the use of the existing method in several rivers in the southern part of Africa.

My earlier suggestion was not corrected - that in the entire manuscript, when the Authors write about the Tsitsa River – the basin or catchment must be added because the readers will have a feeling that the authors studied only this river, not 5 rivers in the Tsitsa River basin, e g., L107, 109 – this correction were incorporated only in lines 1070 and 109 but not in the other parts of the manuscript (e.g. 253, 316 and others)

Discussion is still uncorrected, written in the context of the Tropical zone, African region. For a Peej J as an internationally prestigious journal a wider context is needed and comparison with different rivers in the other parts of the world.

Validity of the findings

Conclusion – this part of the manuscript was not corrected. Authors write that study demonstrated that semi-urban pollution had varying effects on the FFGs- but in fact, these effects are very small. Differences in the taxa abundance in the three types of study sites are small. The composition of fauna is similar, polluted sites have small values of measured parameters, it is strange that from such differences, the statistical significance was obtained. Rivers in the catchment can vary in terms of water flow, bottom sediments, depth, and also other factors that can have an impact on invertebrate assemblages so the results are rather weak.

Additional comments

The manuscript in the present version is not ready for publication in Peer J. Small amount of data was used, and a low number of sampling sites suggest a local problem- I do not say that more samples should be taken and analyzed but at least describe them in a broader context that is still missing. A large part of the comments was omitted.

Reviewer 3 ·

Basic reporting

The revision made by the authors does not seem complete. Even if the authors do not revise the manuscript based on reviewers' comments, the authors should explain the reason.

Experimental design

no comment

Validity of the findings

no comment

Additional comments

[1] In the first comment from the reviewer 3:
"in developing tropical countries." should be " in developing countries." because the authors do not specifically mention tropic countries in this case. "toneonicotinoid" should be "to neonicotinoid".

[2] In the third comment from reviewer 3: the authors do not include the Mzimvubu River in the map, although they mentioned this river in the main text.

[3] In the fourth comment from reviewer 3: I have no idea how the authors revised the manuscript, figures, and tables based on the comment.

[4] In the fourth comment from reviewer 3: I have no idea how the authors revised the manuscript based on the comment.

---

## Round 0.3 · accepted · Accept

Thank you for carefully addressing the concerns expressed in the second round of reviews. The manuscript is now acceptable for publication.